# Epsilon-Globin HBE1 Enhances Radiotherapy Resistance by Down-Regulating BCL11A in Colorectal Cancer Cells

**DOI:** 10.3390/cancers11040498

**Published:** 2019-04-08

**Authors:** Sang Yoon Park, Seon-Jin Lee, Hee Jun Cho, Jong-Tae Kim, Hyang Ran Yoon, Kyung Ho Lee, Bo Yeon Kim, Younghee Lee, Hee Gu Lee

**Affiliations:** 1Immunotherapy Convergence Research Center, Korea Research Institute of Bioscience and Biotechnology, Daejeon 34141, Korea; psy478@kribb.re.kr (S.Y.P.); hjcho@kribb.re.kr (H.J.C.); kjtdna@naver.com (J.-T.K.); yhr1205@kribb.re.kr (H.R.Y.); leekh@kribb.re.kr (K.H.L.); bykim@kribb.re.kr (B.Y.K.); 2Department of Biochemistry, College of Natural Sciences, Chungbuk National University, Cheongju 28644, Korea; yhl4177@chungbuk.ac.kr; 3Environmental Disease Research Center, Korea Research Institute of Bioscience and Biotechnology, Daejeon 34141, Korea; sjlee@kribb.re.kr; 4Department of Biomolecular Science, University of Science and Technology (UST), Daejeon 34141, Korea

**Keywords:** ER stress, HBE1, BCL11A, JNK, radioresistant, oxidative stress, cell cycle arrest

## Abstract

Resistance to radiotherapy is considered an important obstacle in the treatment of colorectal cancer. However, the mechanisms that enable tumor cells to tolerate the effects of radiation remain unclear. Moreover, radiotherapy causes accumulated mutations in transcription factors, which can lead to changes in gene expression and radiosensitivity. This phenomenon reduces the effectiveness of radiation therapy towards cancer cells. In the present study, radiation-resistant (RR) cancer cells were established by sequential radiation exposure, and hemoglobin subunit epsilon 1 (HBE1) was identified as a candidate radiation resistance-associated protein based on RNA-sequencing analysis. Then, compared to radiosensitive (RS) cell lines, the overexpression of HBE1 in RR cell lines was used to measure various forms of radiation-induced cellular damage. Consequently, HBE1-overexpressing cell lines were found to exhibit decreased radiation-induced intracellular reactive oxygen species (ROS) production and cell mortality. Conversely, HBE1 deficiency in RR cell lines increased intracellular ROS production, G2/M arrest, and apoptosis, and decreased clonogenic survival rate. These effects were reversed by the ROS scavenger N-acetyl cysteine. Moreover, HBE1 overexpression was found to attenuate radiation-induced endoplasmic reticulum stress and apoptosis via an inositol-requiring enzyme 1(IRE1)—Jun amino-terminal kinase (JNK) signaling pathway. In addition, increased HBE1 expression induced by γ-irradiation in RS cells attenuated expression of the transcriptional regulator BCL11A, whereas its depletion in RR cells increased BCL11A expression. Collectively, these observations indicate that the expression of HBE1 during radiotherapy might potentiate the survival of radiation-exposed colorectal cancer cells.

## 1. Introduction

Radiotherapy can reduce the risk of cancer recurrence and metastasis. Accordingly, radiation therapy is generally used before and after surgery [1,2]. For colorectal cancer patients, neoadjuvant irradiation has significant benefits compared to surgery alone. After five sessions of short-course radiotherapy at 5 Gy, the overall survival rate has been found to increase from 30% to 38%, and the local recurrence rate decreases from 26% to 9% [3,4]. However, although radiotherapy can represent a considerably effective treatment strategy, the radioresistance of tumor cells often represents a significant obstacle to effective radiation-based therapy [5]. 

Radiation-induced cell death is caused by DNA damage, either directly through the ionization of DNA or indirectly through the generation of reactive oxygen species (ROS) [6,7]. Indirect damage occurs via ROS produced by the radiolysis of water, the principal component of intracellular fluid. Direct or indirect damage to DNA collectively contributes to the DNA damage response (DDR) [8]. Radiation-induced DDR can lead to DNA repair, G2/M arrest, and apoptosis [9,10]. An important determinant of genetic instability and the cellular response to DNA damage is the level of intracellular ROS, which is associated with oxidative stress and can function in cell signaling, as in cell division apoptosis, depending on ROS concentrations [11]. Furthermore, radiotherapy can cause variations in the antioxidant/oxidant system in cancer patients [12,13]. Therefore, antioxidants can be used to prevent the accumulation of ROS, which might improve the survival rate of patients.

Recently, it has been shown that excessive stress, such as that associated with ionizing radiation, can promote endoplasmic reticulum (ER)-related stress and initiate the unfolded protein response (UPR) [14]. Although the UPR is known as a protective response, in cases of excessive or prolonged ER stress, UPR signals can initiate cell death [15]. UPR signaling is initiated via three distinct ER-resident protein folding sensors, namely, inositol-requiring enzyme 1 (IRE1), protein kinase RNA-like ER kinase (PERK), and activating transcription factor 6 (ATF6) [16]. Radiation-induced ER-mediated cell death has been linked to oxidative stress caused by the overproduction and accumulation of ROS [17]. IRE1a is a transmembrane protein composed of an ER lumenal domain that is oligomerized during ER stress [18]. IRE1a plays at least two different roles under conditions of ER stress. First, in response to ER stress, X-box binding protein 1 (XBP-1) mRNA is converted to its potent transcriptional activator XBP1S by IRE1. This cleavage affects cell homeostasis [19]. Second, under excessive ER stress conditions, prolonged activation of IRE1 results in its interaction with tumor necrosis factor receptor (TNFR)-associated factor-2 and ultimately leads to the phosphorylation of Jun amino-terminal kinase (JNK) [20]. Activation of JNK in turn enhances downstream responses and eventually initiates apoptosis [21].

By comparing RNA-seq data from several colorectal cancer (CRC) cell lines with differing radiation sensitivities, we identified a candidate radiation resistance-associated gene, namely, epsilon-globin (HBE1). The human β-like globin, located on chromosome 11, consists of five homologous genes (5′-ε−Gγ-Aγ-δ-β−3′), which are expressed in a strictly controlled sequence during development. This process is referred to as hemoglobin switching [22]. Although in adults, the expression of HBE1 is normally silenced, the encoding gene remains structurally intact [23]. However, recent studies have shown that reactivation of epsilon-globin gene expression in mouse models ameliorates sickle cell disease and β-thalassemia [24]. Furthermore, in contrast to alpha-globin, one of the beta-like globins, beta-globin (HBB), has been found to be more consistently up-regulated in circulating tumor cells. This tumor-specific induction of HBB leads to the suppression of intracellular ROS levels in cancer cells. Consequently, the expression of HBB enhances metastatic ability in mouse models [25]. Surprisingly, HBE1 is also markedly overexpressed in lymph node metastatic variant CRC cell lines [26].

In this study, we established radioresistant cell lines and used these cells to examine the radiation-induced genetic changes associated with radiation resistance and to gain insights into the underlying molecular mechanisms. We identified an increase in the expression of HBE1 in radioresistant cells, and accordingly believe that understanding the precise molecular mechanisms associated with epsilon-globin gene switching will provide a potential strategy for targeted radiation therapy.

## 2. Results

### 2.1. Establishment and Characterization of Radioresistant CRC Cell Lines

To identify the genes associated with radiation-induced cell damage, we initially established radioresistant CRC cell lines. CRC cells were exposed to ionizing radiation over several weeks for a total exposure of 120 Gy. Thereafter, we selected the surviving cells. These surviving cells were characterized by radiation resistance that was lacking in parental cells. Subsequently, we further investigated whether the SW480-IR and HT-29-IR lines showed resistance to radiation-induced cell death compared to the parental cells. To substantiate the resistance to radiation, we performed a clonogenic assay. Compared to that in the parental cells, the ionizing radiation-resistant cell lines showed an increased survival rate (Figure 1A). We also monitored the degree of apoptosis following irradiation. Cells were irradiated with 5 Gy radiation, and then 48 h later, they were stained with annexin V and analyzed using a FACSverse flow cytometer (Figure 1B). Results indicated that the ionizing radiation-resistant cells showed decreased radiation-induced apoptotic cell death. Given that the amplification of cleaved caspase proteins after exposure to irradiation is known to stimulate cell death [27], we evaluated whether apoptosis was caspase-dependent by performing western blot analysis. As shown in Figure 1C, 48 h after exposure to 5 Gy of ionizing radiation, HT-29 and SW480 cells showed significantly increased levels of caspase-3, caspase-7, caspase-9, and PARP cleavage compared to those in respective ionizing radiation-resistant cell lines. Cellular checkpoints have been reported to be among the various factors that influence the effect of radiation. In particular, the G2/M checkpoint is recognized as an indicator of radiosensitive cell cycle status. Unlike the G1/S checkpoint, this checkpoint is rapidly activated in response to irradiation-induced damage [28]. Moreover, radiation-induced delayed apoptosis occurs during the G2/M phase and mitotic cell death is linked to incomplete mitosis [29]. We confirmed radiation dose-dependent cell cycle distribution. Both SW480-IR and HT-29-IR cells showed the reduced accumulation of cells in the G2/M phase (Figure 1D). Thus, our results indicated that SW480-IR and HT-29-IR cells had acquired greater resistance to radiation-induced cell death.

### 2.2. HBE1 Enhances the Radiation Resistance of CRC Cell Lines

To identify candidate proteins involved in radiation resistance, we used RNA-seq technology to detect differences between radiation-resistant and parental cells; we then examined the differential expression of *HBE1* mRNA using RT-PCR. When we analyzed the expression levels of basal *HBE1* mRNA in CRC cells (SW480, SW480-IR, SW620, SW620-IR, HT-29, HT-29-IR, RKO, and RKO-IR cells), we found that it was approximately 6-fold higher in radioresistant cells (SW480-IR, SW620-IR, HT-29-IR, and RKO-IR) than in respective radiosensitive cells (SW480, SW620, HT-29, and RKO cells) (Figure 2A, upper). We also demonstrated that, following transient radiation treatment (120 Gy), the CRC cell lines HCT-116, DLD-1, KM12C, and CACO-2 showed increased *HBE1* mRNA levels compared to those in control cells (Figure 2A, lower). These findings led us to hypothesize that HBE1 expression might be related to radiation resistance in radioresistant CRC cells. To examine whether HBE1 expression affects radiation resistance, we transfected CACO-2, HCT-116, DLD-1, and KM12C CRC cells with a pCMV6-HBE1 vector system, using cells transfected with the pCMV6 vector as controls. The results of a clonogenic assay using cells exposed to various doses of radiation revealed that those cells characterized by HBE1 overexpression had an increased survival rate, thereby indicating that this protein might play a functional role in enhancing radiation resistance (Figure 2B). In contrast, when we knocked down HBE1 in the radioresistant cell lines SW480-IR, SW620-IR, HT-29-IR, and RKO-IR, via HBE1-specific siRNA, we found that HBE1 depletion significantly reduced radiation-induced cell growth inhibition (Figure 2C). To assess the effect of radiation on the cell cycle, we also examined the effect of HBE1 on radiation-induced G2/M accumulation following exposure to 5 Gy radiation. As shown in (Figure 2D), radiation-induced G2/M accumulation was reduced by the overexpression of HBE1 in SW480 and HT29 cells. We confirmed that the radiation-induced cell cycle distribution in HBE1-overexpressing cells showed a similar pattern to that in radiation-resistant cell lines. As a result, HBE1 expression was thought to be involved in G2/M cell cycle transition.

### 2.3. Radiation-Induced ROS Generation is Regulated by HBE1 Expression

Radiation-induced cellular damage is largely mediated by the generation of ROS. The production of ROS plays a critical role in genomic instability, which can lead to ionizing radiation-induced cytotoxicity. We thus examined the levels of ROS in CRC cells using DCFDA staining after exposure to ionizing radiation. In response to exposure to 5 Gy of ionizing radiation for 24 and 48 h, we found that ROS generation in SW480 and HT29 cells was markedly increased compared to that in the corresponding radiation-resistant cell lines (Figure 3A). In an effort to determine whether HBE1 expression is associated with the radiation resistance properties of CRC cells, we established HBE1-overexpressing cell lines from SW480 and HT29 cells and examined radiation-induced apoptosis by annexin V and PI staining. Figure 3B shows that the number of apoptotic cells induced by 5 Gy radiation in HBE1-overexpressing cells was lower than that in parental cells, thereby indicating that the overexpression of HBE1 reduces radiation-induced cell death. The increased radiation resistance detected in CRC cells raises the possibility that the expression of HBE1 is linked to elevated levels of intracellular ROS. Upon exposing SW480 cells to hydrogen peroxide, which increases intracellular ROS, we found that it could not promote a marked dose-dependent induction of *HBE1* mRNA (Figure 3C). To investigate how HBE1 expression modulates radiation-induced intracellular ROS generation, we exposed mock and HBE1-overexpressing SW480 and HT-29 cells to radiation in a time-dependent manner and measured the level of intracellular ROS production (Figure 3D). The results of clonogenic assays indicated that HBE1-overexpressing cells exhibited increased survival compared to that in control cells following radiation exposure, and that the effect could be suppressed by pre-treatment with the ROS scavenger NAC (Figure 3E). Likewise, radiation-induced ROS production was restored by NAC treatment (Appendix A). These data revealed that the overexpression of HBE1 can attenuate radiation-induced ROS production in both SW480 and HT-29 cell lines, thereby indicating that HBE1 has a protective effect against intracellular oxidative stress following exposure to radiation. 

### 2.4. Enrichment of HBE1 Levels Decreases Radiation-Induced ER-Stress Signaling

Cells experience multiple cytotoxic stresses after exposure to ionizing radiation. Thus, various signaling events can lead to reprograming after several exposures to radiation. To determine the altered cellular signaling pathways in HBE1-expressing cells, we performed western blotting and examined whether down-regulation of the pro-apoptotic pathway contributed to a decrease in radiation-induced cell death. The irradiation of cells led to an increase in JNK activity within 60 min and the effect persisted for up to three days. Compared to that in control cells, cells expressing HBE1 showed reduced p-JNK levels (Figure 4A). Given that JNK activates apoptotic signaling pathways, these results indicate that HBE1-expressing CRC cells acquire radioresistance. Furthermore, radiation-induced ER stress can lead to the activation of JNK, which in turn can trigger apoptosis in cancer cells. To determine the pathway through which HBE1 inhibits radiation-induced ER stress, we performed western blotting after exposing cells to 5 Gy of ionizing radiation in a time-dependent manner (Figure 4B). The results showed that the expression level of p-IRE1a was significantly increased in parental cells after one day of exposure to 5 Gy radiation, whereas in HBE1-expressing cells, IRE1a phosphorylation was delayed and reduced. In addition, the IRE1a-mediated splicing of *XBP1* mRNA, XBP1(S), decreased in HBE1-expressing cells (Appendix A). However, in both cell lines examined, there were no appreciable differences in the levels of p-eif2a. Activation of the eif2a–CHOP pathway results in the up-regulation of ATGs, whereas the IRE1 pathway elicits apoptosis via a JNK-mediated signaling pathway [30]. Thus, we assume that the expression of HBE1 protects CRC cells from cell death induced by radiation. In addition, treatment with the JNK inhibitor SP600125 was observed to decrease radiation-induced cell death in a concentration-dependent manner (Figure 4C). To further investigate whether the generation of ROS affects ER stress in response to radiation exposure, we examined the effect of NAC on the level of p-IRE1a under conditions of radiation-induced ER stress. We accordingly detected the phosphorylation of IRE1a and JNK in mock and HBE1-expressing cells exposed to 5 Gy radiation. In contrast, combined treatment with NAC and radiation led to a decrease in p-IRE1a and p-JNK levels in both cell types (Figure 4D). In particular, compared to that in mock cells, we observed a significant reduction in the enhancement of radiation-induced ER stress mediated by NAC treatment in HBE1-expressing cells. Our results thus indicate that ER stress caused by the generation of ROS appears to be correlated with the absence of HBE1 in CRC cells.

### 2.5. HBE1 Expression Levels Are Regulated by the Depletion of BCL11A

The accumulation of mutations in transcription factors can lead to changes in gene expression in radiation-exposed cells [31]. A recent study indicated that HBE1 expression might be regulated by transcription factors such as NR2C1, NR2C2, MBD2, YY1, SOX6, and BLC11A, the last of which is reported to be an HBE1 suppressor and was selected as a significantly down-regulated transcription factor [32,33,34,35,36]. As shown in Figure 5A,B, BLC11A levels were reduced in radiation-resistant cells, thereby indicating that BCL11A expression level might negatively influence *HBE1* mRNA expression in CRC cells. To confirm whether HBE1 expression is regulated by BCL11A levels, we constructed a flag-tagged BCL11A vector and used this to transfect SW480 cells. qPCR analysis data showed that the induction of BCL11A attenuates *HBE1* mRNA levels in a concentration-dependent manner (Figure 5C). In addition, depletion of BCL11A in radiation-resistant cells via siRNA elevated the expression of HBE1 with or without radiation exposure (Figure 5D). Interestingly, a determination of cell survival after exposure to gamma irradiation using a clonogenic assay indicated that survival rates were also effectively enhanced by BLC11A knockdown. In contrast, the transfection of cells with a negative control vector did not affect radiation-induced cell death (Figure 5E). Collectively, these data indicate that BLC11A-mediated *HBE1* mRNA expression levels affect the radiation resistance of CRC cells.

### 2.6. HBE1 Deficiency Affects Radiation Sensitivity

Extending our observations regarding the regulation of HBE1 expression with respect to radiation-induced resistance, we performed loss-of-function experiments using siRNA against HBE1 to examine the effects on ROS generation, radiation-induced apoptosis, and cell cycle distribution in radiation-resistant cells. To investigate whether HBE1 depletion promotes an increase in intracellular ROS levels in radiation-resistant CRC cells, we transfected these cells with negative control or HBE1-targeting siRNA and exposed these cells to ionizing radiation. We accordingly observed that the knockdown of endogenous HBE1 in ionizing radiation-resistant cells significantly elevated intracellular ROS levels in a time-dependent manner (Figure 6A). After transfecting radiation-resistant cells (SW480-IR) with HBE1-targeting siRNA, we performed a clonogenic assay on cells with and without pre-treatment with NAC and observed the survival rates of cells with HBE1 knockdown. Cell death was increased compared to that in cells transfected with control siRNA and survival rates in the latter condition were recovered with NAC treatment (Figure 6B). Moreover, a measurement of radiation-induced apoptosis rates by flow cytometry revealed increased apoptosis following HBE1 knockdown, consistent with our previously obtained results indicating that HBE1 contributes to cellular radiation resistance (Figure 6C). Furthermore, we observed that cell cycle arrest at G2/M was increased by HBE1 knockdown in a dose-dependent manner in both SW480-IR and HT-29-IR cell lines (Figure 6D). These results thus indicate a correlation between increased ER stress-mediated apoptosis and G2/M cell cycle arrest. Taken together, our results indicate that elevated HBE1 expression is linked to radiation resistance in CRC cells via enhanced cellular defenses against ROS-related stress.

## 3. Discussion

Depending on the different degrees of radiation sensitivity or resistance, radiotherapy induces a complex response in various types of tumors, such as breast, gastric, and head and neck squamous carcinoma cells [37,38,39]. In response to radiation, certain cellular processes are initiated to repair damage, whereas others induce cell death [40]. The repair of radiation-induced damage is crucial to genomic integrity, and deficiencies in the molecules involved in repair processes are known to be associated with cell mortality following ionizing radiation. Recent studies have indicated that aberrantly activated signal transduction pathways in cancer cells can influence cellular radiation sensitivity [41]. In the present study, we generated radiation-resistant cell lines, which are a good model to examine the different responses to radiation, and identified novel candidate molecules implicated in radiation resistance using RNA-sequencing analysis. We accordingly revealed that the transcriptional levels of epsilon-globin (HBE1) in radiation-resistant CRC cell lines were specifically associated with resistance to radiation, and that this protein has a protective effect against various forms of radiation-induced cellular damage, including G2/M accumulation, ROS generation, and apoptosis. 

Hemoglobin is generally considered to be exclusively present in cells and the expression of its α and β chains is tightly coordinated. The physiological α2β2 hemoglobin complex binds oxygen in pulmonary capillaries and releases it in distal tissues; however, the abnormal β4 hemoglobin H tetramer found in patients with alpha-thalassemia is unstable and has a higher affinity for oxygen [42]. The expression of globin chains has also been reported in a number of non-erythroid tissues including neuronal cells, macrophages, and hepatocytes, as well as in several types of epithelial cancers, such as thyroid and breast tumors [43,44]. Although the functional significance of globin chain expression remains uncertain, it has been shown to promote oxygen storage in neurons and to reduce oxidative stress in mesangial cells [45], and the antioxidant function of hemoglobin B was found to contribute to the survival of cancer cells [25]. These findings indicate that HBE1 might play an antioxidant-related role against radiation-induced oxidative stress. In the present study, we observed that cells overexpressing HBE1 are characterized by reduced intracellular ROS generation when exposed to radiation. Consequently, radiation-induced cell death is likewise reduced. Furthermore, we found that in ionizing radiation-resistant cell lines, ER stress-induced JNK activation is impaired. Under severe ER stress conditions, the IRE1a–JNK signaling pathway induces apoptosis [46]. A reduction in ROS generation in HBE1-expressing cells exposed to radiation reduces ER stress-induced apoptosis, which is reversed by treatment with N-acetyl cysteine. Interestingly, the depletion of HBE1 in ionizing radiation-resistant cells has contrasting effects, increasing ROS generation, G2/M arrest, and apoptosis and reducing the clonogenic survival rate following radiation exposure.

Radiation exposure rapidly leads to the generation of cellular ROS, notably hydroxyl radicals, as well as the less-investigated reductants hydrogen radical, hydrated electrons, superoxide, and hydrogen peroxide, which are formed as secondary ROS products that are induced by ionizing radiation [47]. Exposure to ionizing radiation has been definitively linked to mitochondrial-dependent ROS/reactive nitrogen species generation in tumor cells [48], and increased ROS generation in mitochondria following low-dose ionizing radiation has been shown to contribute significantly to radiosensitivity and cell death. [49] Moreover, a decline in tissue antioxidants is observed during radiotherapy for breast cancer patients [50] and an aberrant redox status affects the host immune system [51]. However, there is some debate regarding whether the use of antioxidants is beneficial or detrimental [52]. Nevertheless, antioxidants are considered an important factor in the response of tumors to radiation therapy [53]. These observations are consistent with the concept that the expression levels of hemoglobin subunits might in fact represent a more important response to radiotherapy in CRC cells.

The autonomous silencing of the epsilon-globin gene is mediated by several different known factors, which bind to disparate sites near the coding sequences of the gene; in addition, several transcription factors including NR2C1, NR2C2, MBD2, YY1, SOX6, and BLC11A have been reported to directly regulate epsilon-globin gene expression [32,33,34,35,36]. In addition, the epigenetic modulator SUV-20h1 and histone lysine methyltransferase have been shown to alter epsilon-globin expression [24]. To assess the utility of this system, we performed RNA interference (RNAi)-mediated knockdown of BCL11A in the presence or absence of known fetal hemoglobin inducers. Our results revealed increased epsilon-globin mRNA levels in the radiation-resistant environment following BCL11A depletion. Furthermore, disruption of the erythroid-specific enhancer of BCL11A decreased oxidative damage via reduced ROS generation [54]. Furthermore, BCL11A is known as a non-autonomous T cell tumor suppressor [55]. Interestingly, another BCL11 family member, BCL11bB, acts as a tumor suppressor. Therefore, disruption of BCL11B contributes to oncogenesis. Moreover, BCL11B-lacking cells were found to exhibit deregulation of the cell cycle checkpoint [56]. Inappropriate cell cycle progression induces the accumulation of DNA damage and ultimately contributes to tumor progression. In contrast, the down-regulation of BCL11A is known to induce apoptosis in B lymphoma cells [57] and BCL11A has also been reported to be associated with a variety of B-cell malignancies in humans [58]. These results make it difficult to distinguish whether BCL11A is an oncogene or a tumor suppressor gene. Thus, further investigation of BCL11A function is required for effective radiotherapy strategies.

Based on these results, we can infer that the expression of HBE1 is altered by an external factor such as radiation or epigenetic modification. Ultimately, further biochemical studies will be required to characterize the physiological properties of epsilon-globin as a factor implicated in the radiation sensitivity and resistance of CRC cells. 

## 4. Materials and Methods

### 4.1. Cell Culture

The colorectal cancer cell lines SW480, SW620, HT-29, and RKO were purchased from the Korean Cell Line Bank (SNU, Seoul, Korea) and cultured in DMEM (Gibco, Waltham, MA, USA) supplemented with 10% fetal bovine serum (HyClone, Logan, UT, USA) and antibiotics (100 U/mL penicillin and 100 μg/mL streptomycin (Gibco-BRL)) at 37 °C in a humidified incubator with 5% CO_2_. N-acetyl-cysteine (NAC) was purchased from Sigma-aldrich (St. Louis, MO, USA). The following antibodies were purchased by Santa Cruz Biotechnology (Santa Cruz, CA, USA): anti-GRP78, anti-CHOP, phospho-IREa, phospho-eIF2a, and anti-β-actin. The following antibodies were obtained from Cell signaling Technology (Danvers, MA, USA): anti-cleaved Caspase-3, -7, -9, anti-cleaved PARP, anti-phospho-JNK, and anti-JNK antibody. The JNK inhibitor SP600125 was purchased from Enzo Life Sciences (Farmingdale, NY, USA). The MODEL 109 IRRADIATOR (JL Shepherd and Associates) was used to generate γ-rays for in vitro experiments.

### 4.2. Irradiation

Colorectal cancer cells were plated in 60 mm dishes and incubated at 37 °C in a humidified, 5% CO2/air atmosphere. After 24 h, cells were exposed to irradiation from 60CO source at the indicated dose of Gy/minute. For generation of radiation resistance colorectal cancer cell lines using SW480, HT-29, SW620, and RKO cell lines, the radiation resistant cancer cells were established by treatment with parental cells. CRC cells were plated in 60mm dishes at a density of 5 × 10^3^ cells/dish and exposed to 5 Gy dose of ionizing radiation, followed by 15 days recovery. This process repeated for 24 treatments total 120 Gy.

### 4.3. Construction of the HBE1 Expression Plasmid and Transfection

Human HBE1 cDNA (NM_005330) was obtained from ORIGENE (RC02247). HBE1 expression vector was transfected using lipofectamine 2000 reagent (invitorgen), according to the manufacturer’s protocol and recovered in RPMI1640 medium (Welgene) containing 10% fetal bovine serum for 24 hrs. After recovering, viable cells were calculated by WST-1. HBE1 stable overexpressed cell line was constructed by using pCMV6-HBE1 plasmid. In stable overexpressed cell lines, SW480 and HT-29 cells (1 × 105 cells/well) were seeded into 24-well plate. After 16 h, HBE1-expressed vector was transfected into SW480 and HT-29 cells using lipofectamine 2000 (Invitrogen, Carlsbad, CA, USA). The following day, the medium was changed, and G418 (Calbiochem, Billerica, MA, USA) was added to the culture medium to a final concentration of 800 μg/mL and cultured in the presence of G418 for 4 weeks. Medium was exchanged every 3 days. The expression of HBE1 to identify establishment of HBE1 stable cell line was checked by real time-PCR, western blot analysis, and clonogenic ssay.

### 4.4. RNA Interference Experiments

siRNA duplexes of HBE1 or BCL11A were purchased from Bioneer (Daejeon, Korea). The specific target sequence of HBE1 siRNA was sense 5’-GAG AAG GCU GCC GUC ACU A dTdT-3’ antisense 5’-UAG UGA CGG CAG CCU UCU CdTdT-3 and the specific target sequence of BCL11A siRNA was sense 5’-CGA UUG UUU AUC AAC GUC AdTdT-3’ antisense 5’-UGA CGU UGA UAA ACA AUC GdTdT-3. siRNA duplexes were transfected using lipofectamine RNAiMAX reagent (Invitrogen). 

### 4.5. Western Blot Analysis

Cancer cell lines were solubilized in radio-immunoprecipitation assay (RIPA) lysis buffer (50 mM Tris-HCl (pH 7.4), 150 mM NaCl, 1% NP40, 0.25% sodium deoxycholate, 1 mM phenylmethylsulfonylfluoride (PMSF), 1 mM sodium orthovanadate, 1× sigma protease inhibitor cocktail) and protein was measured using a standard bicinchoninic acid assay. Equal amounts of protein (20–50 μg) were size-fractionated by 10%~15% SDS-PAGE and then transferred onto PVDF membrane (Millipore Corporation, Billerica, MA, USA). Membranes were blocked by incubation for 1 h with 5% skim milk/PBS-T buffer (PBS with 5% powdered milk and 1% Triton X-100), and incubated overnight at 4 °C with primary antibodies diluted in 1× PBST buffer. The following primary antibodies were used and the membranes were washed 3 times with PBST. Secondary antibodies were diluted in PBST and were added for 40 min at room temperature. The following secondary antibodies were used and membranes were washed 6 times with PBST for 1 h. The blots were visualized by chemiluminescence (Clarity Western ECL; Bio-rad, Hercules, CA, USA).

### 4.6. Flow Cytometry for Measurement of ROS Generation and Apoptosis Assay

The fluorescent probe CM-H2DCFDA was used for the assessment of intracellular level of ROS. Briefly, cells were exposed to 5 Gy radiation and incubated for 24 h. After exposure, cells were stained with 10 μM H2DCFDA at 37 °C for 15 min in the dark. The stained cells were analyzed with FACS Calibur flow cytometer (Becton Dickinson, San Jose, CA, USA). For apoptosis analysis, cells were harvested and centrifuged at 800 rpm for 3 min after irradiation (48 h). Cells were carefully resuspended and added in annexin-V (5 μL) and propidium iodide (PI) (5 μL) (BD biosciences, San Jose, CA, USA) then incubated at 37 °C for 15 min in the dark. The stained cells were analyzed using a FACSverse flow cytometer (BD Bioscience).

### 4.7. Clonogenic Assay

Cells were seeded at 1000, 2000, 5000, 8000, and 10,000 cells per 60 mm dish plate. After 24 h, cells were irradiated with 0, 2, 4, 6, and 8 Gy dose of ionizing radiation, respectively. Then cells were incubated for 14 days at 37 °C. Colonies were fixed and stained with 0.1% crystal violet in 20% ethanol and counted. Survival fraction was expressed as the relative plating efficiencies of the irradiated cells compared with that of the non-irradiated cells. 

### 4.8. Cell Cycle Analysis

Cells were irradiated with 0, 2, and 4 Gy dose of ionizing radiation, respectively. Then, 24 h later, irradiated Cells were trypsinized, washed in PBS, and fixed in 70% ethanol for 30 min on ice. Fixed cells were suspended in PI (BD biosciences) containing 0.1 mg/mL RNase A (BIOFACT, Daejeon, Korea); then, incubated at 37 °C for 15 min. DNA content was analyzed using a FACSCalibur flow cytometer (BD Bio-science) and the Flowjo software v7.6.1. (Flowjo, Ashland, OR, USA). At least 10,000 cells per replication were analyzed and two replicas were performed at each time point.

### 4.9. Real-Time Quantitative PCR

Total RNA was prepared using Trizol (Life Technologies, Life Technologies, Grand Island, NY, USA) according to the manufacturer’s instruction. Reverse transcription was conducted using 10 µg of total RNA with a reverse transcription kit (Promega, Madison, WI, USA). A total of 1 μL of cDNA was used for the PCR, and triplicate reactions were performed for each sample using a Power SYBR Green Kit (Applied Biosystems, Foster City, CA, USA) with gene-specific primers on an ABI StepOnePlus instrument (Applied Biosystems). The PCR primers were as follows: HBE1, sense 5’-GCA AGA AGG TGC TGA CTT CC-3’ antisense 5’-TGC CAA AGT GAG TAG CCA GA-3’; BCL11A, sense 5’-GCC AGA GGA TGA CGA TTG TT-3’ antisense 5’-GCT GCT GGG CTC ATC TTT AC-3’; b-actin, sense 5’-AAG GCC AAC CGC GAG AAG AT-3’ antisense 5’-TGA TGA CCT GGC CGT CAG G-3’. 

RNA quantity was normalized to β-actin content, and gene expression was quantified according to the 2^−ΔCt^ method.

### 4.10. Reverse-Transcription PCR

Reverse-transcription PCR was performed using 2× TOPsimpleTM DyeMIX-Tenuto (Daejeon, KOREA) on a ProFlex PCR system (Applied Biosystems). The gene-specific PCR primers were as follows: XBP1, sense 5’- CCT TGT AGT TGA GAA CCA GG-3’ antisense 5’-GGG GCT TGG TAT ATA TGT GG-3’; CHOP, sense 5’-GCA CCT CCC AGA GCC CTC ACT CTC C -3’ antisense 5’- GTC TAC TCC AAG CCT TCC CCC TGC G-3’; HBE1, sense 5’-TTT GGA AAC CTG TCG TCT CC-3’ antisense 5’-CCT TGC CAA AGT GAG TAG CC-3’; GAPDH, sense 5’-CGA GAT CCC TCC AAA ATC AA-3’ antisense 5’-TGT GGT CAT GAG TCC TTC CA-3’. Each PCR product was electrophoresed on 2% agarose gels. GAPDH served as the standard.

### 4.11. Statistical Analysis

All results were confirmed in at least three independent experiments; data from one representative experiment are shown. All quantitative data are presented as mean ± standard deviation (SD); in vivo data are expressed as mean ± standard error of the mean (SEM). Statistical analysis was performed using SAS 9.2 software (SAS Institute, Cary, NC, USA). Student’s *t*-tests were used for comparisons of means of quantitative data between groups and *p* < 0.05 was considered statistically significant.

## 5. Conclusions

The expression of HBE1 might lead to the attenuation of radiation-induced cell death, based on its effects on radiation resistance and ROS generation, which might not be evident by analyzing established primary or metastatic tumor specimens. It is noteworthy that the marked overexpression of HBE1, evident in radiation-resistant CRC cells, does not occur in either primary tumors or in metastatic tumors, but rather reflects the transient state of cancer cells in the vasculature. In conclusion, we suggest that HBE1 could be a potential antioxidant and that its expression in response to radiation might indicate the acquisition of radiation resistance. Therefore, factors contributing to the expression of HBE1 could represent potential molecular targets to enhance the efficacy of radiation therapy.

## Figures and Tables

**Figure 1 cancers-11-00498-f001:**
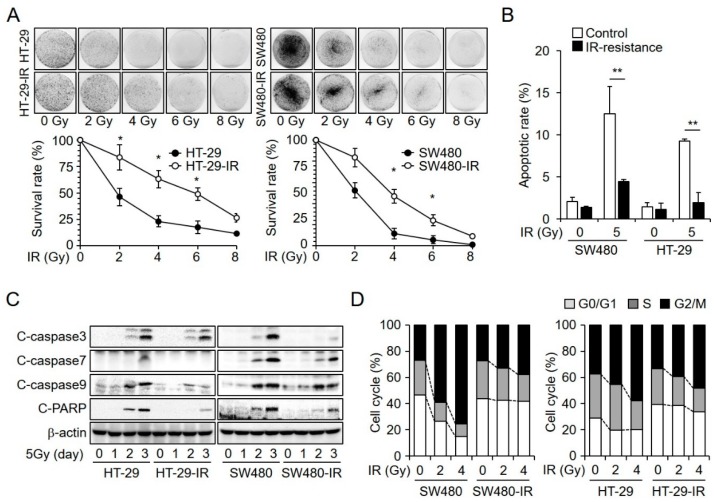
Establishment and characterization of irradiation-resistant colorectal cancer cell lines. (**A**) A clonogenic assay revealed that the radiation-resistant cell lines HT-29-IR and SW480-IR showed decreased radiation-induced cell mortality compared to that in the radiosensitive parental cell lines. Both sensitive and resistant cell lines were exposed to 0, 2, 4, 6, and 8 Gy gamma-irradiation. (* *p* < 0.05). (**B**) Analysis of apoptosis by flow cytometry in radiosensitive and radioresistant cell lines. After exposure to 5 Gy of ionizing radiation for 48 h, comparing it to that in parental cells (** *p* < 0.05). (**C**) Western blot analyses were performed to determine the expression levels of cleaved (**C**) caspase 3, 7, and 9 and cleaved PARP. β-actin was used as an internal control. Parental cells were more sensitive to radiation than the corresponding ionization radiation-resistant cells. (**D**) Cell cycle distribution of irradiation-exposed cells (SW480, HT-29, SW480-IR, and HT-29-IR cells) for the indicated doses (0, 2, and 4 Gy). Flow cytometry was used to measure cell cycle arrest.

**Figure 2 cancers-11-00498-f002:**
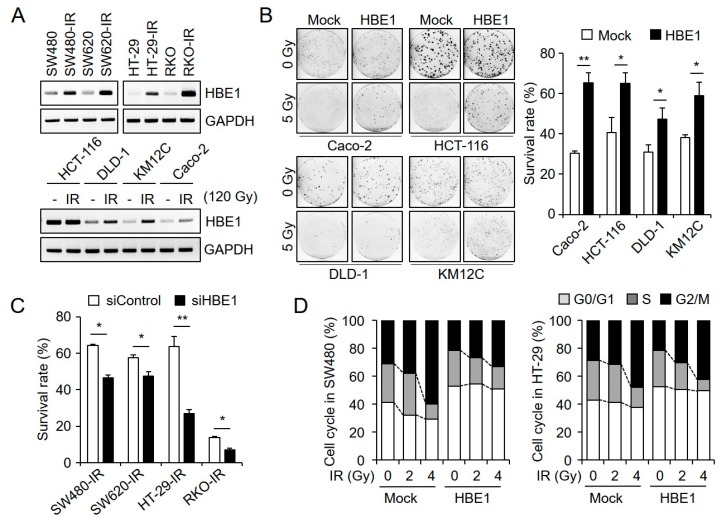
Hemoglobin subunit epsilon 1 (HBE1) expression levels are positively related to radiation resistance in radiation-resistant colorectal cancer cells lines. (**A**) Basal *HBE1* mRNA levels in the radiation-resistant cell lines SW480-IR, SW620-IR, HT-29-IR, and RKO-IR cells were significantly up-regulated compared to those in radiosensitive cell lines (SW480, SW620, HT-29, and RKO cells) (upper panel) as determined by RT-PCR. Colorectal cancer cell lines HCT-116, DLD-1, KM12C, and CACO-2 showed transiently increased *HBE1* transcriptional levels in response to radiation treatment (120 Gy). (lower panel). (**B**) After transiently transfecting colorectal cancer cells with a mock or HBE1-overexpression vector, cells were analyzed by clonogenic assays. The colony-formation ability of HBE1-overexpressing cells was significantly greater than that in mock cells after irradiation. Representative images are shown in the left panel and a quantitative graph is shown in the right panel. * *p* < 0.05, ** *p* < 0.01. (**C**) After transfection with control or HBE1 siRNA in the presence of radiation (5 Gy) exposure, the deficiency in HBE1 decreased the cell survival rate compared to that in control cells, as determined by a clonogenic assay. * *p* < 0.05, ** *p* < 0.01. (**D**) HBE1 overexpression in SW480 (left panel) and HT-29 (right panel) cells attenuated G2/M phase arrest compared to that in mock-treated cells with the indicated levels of irradiation exposure. Graphs present the percentage of cell distribution.

**Figure 3 cancers-11-00498-f003:**
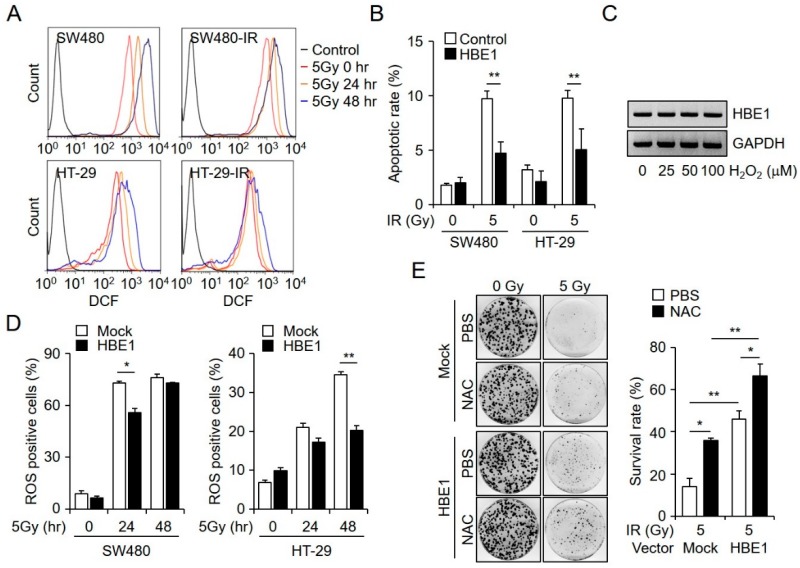
HBE1 expression inhibits radiation-induced cell death by attenuating reactive oxygen species (ROS) generation in colorectal cancer cells. (**A**) CM-H2DCFDA was used to detect radiation-induced ROS generation by flow cytometry at 24 and 48 h after administering 5 Gy of irradiation. (**B**) After the stable transfection of SW480 and HT-29 cells with a control or pCMV-HBE1-overexpression vector, radiation-induced cell death was measured by flow cytometry. After 48 h, both cell lines were irradiated with 5 Gy and apoptotic rates were analyzed by annexin V and PI staining. * *p* < 0.05, ** *p* < 0.01. (**C**) RT-PCR data showing the dose-dependent induction of *HBE1B* mRNA levels in H_2_O_2_-treated SW480 cells. (**D**) Intracellular ROS levels in HBE1-overexpressing SW480 and HT-29 based on CM-H2DCFDA and as measured by flow cytometry, at 24 and 48 h after exposure to 5 Gy of irradiation. * *p* < 0.05, ** *p* < 0.01. (**E**) Following the transfection of SW480 cells with a mock or HBE1-overexpression vector, a clonogenic assay was performed after treating the cells with N-acetyl cysteine (NAC) (10 mM) and exposure to 5 Gy irradiation. The left panel shows a representative clonogenic plate and the right panel depicts the survival rate. All data shown are the means ± SD of three independent experiments. * *p* < 0.05, ** *p* < 0.01 versus the control.

**Figure 4 cancers-11-00498-f004:**
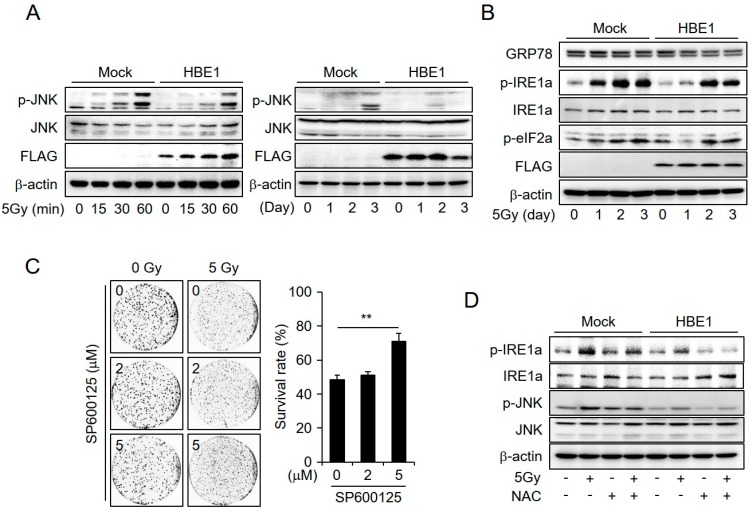
Enrichment of HBE1 levels decreases the radiation-induced activation of Jun amino-terminal kinase (JNK) and endoplasmic reticulum (ER) stress. After the transfection of SW480 cells with a flag-tagged HBE1-overexpression or mock vector in the presence or absence of radiation (5 Gy), the activation of JNK, ER stress-related proteins, and colony formation ability were measured by immunoblot analysis and a clonogenic assay. (**A**) Lysates were analyzed by western blotting to determine the activation level of p-JNK, total JNK, and FLAG for early (left panel) and late (right panel) timepoints after radiation exposure (5 Gy). (**B**) The activation of ER stress signaling proteins was assessed. β-actin was used as a loading control. (**C**) Following exposure to irradiation (5 Gy) in the presence or absence of the JNK inhibitor SP600125 for the indicated doses, cell survival rate was measured using a clonogenic assay ** *p* < 0.01, and compared to that in untreated cells. (**D**) After transfection with an HBE1-overexpression or mock vector in the presence or absence of radiation (5 Gy) exposure and ROS scavenger treatment, the inhibition of radiation-induced ER-stress by N-acetyl cysteine (NAC; 10 mM) was analyzed by western blotting. β-actin was used as a loading control. The phosphorylation of IRE1a and JNK was measured by pretreating cells with NAC before exposure to 5 Gy of irradiation.

**Figure 5 cancers-11-00498-f005:**
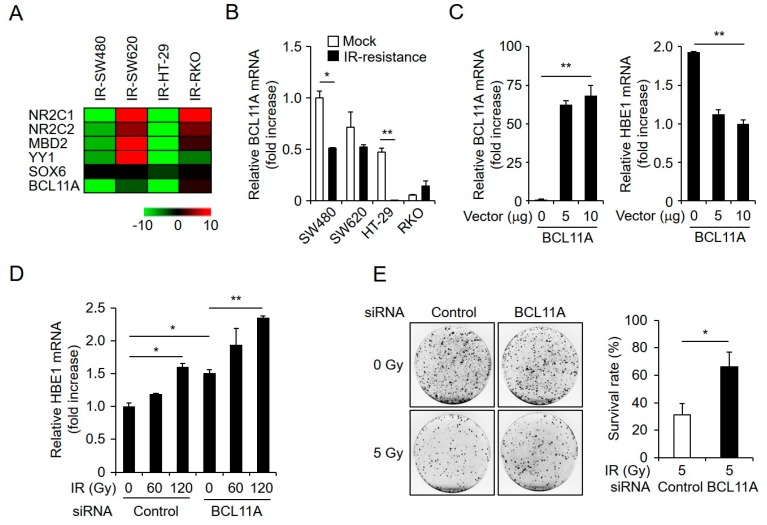
HBE1 expression is negatively regulated by the depletion of BCL11A expression. (**A**) Heatmap of differentially expressed HBE1 suppressor candidate genes (**B**) qRT-PCR confirmation of a candidate HBE1 suppressor gene selected based on RNA-seq analysis. * *p* < 0.05, ** *p* < 0.01. (**C**) Cells were transfected with a pCMV6-BCL11A-overexpression vector in a vector DNA concentration-dependent manner and changes in the mRNA expression level of *BCL11A* (left) and *HBE1* (right) were analyzed by qRT-PCR. ** *p* < 0.01 compared to control cells. (**D**) Cells were transfected with control or BCL11A siRNA and exposed to gamma-irradiation (60 and 120 Gy). *HBE1* mRNA levels were assessed by qRT-PCR. * *p* < 0.05, ** *p* < 0.01 compared to non-irradiated cells. (**E**) Representative image of clonogenic assay cells stained with crystal violet (left panel) and a graph showing mean survival rate as a function of radiation for negative control- or siBCL11A-treated cells. * *p* < 0.05 compared to control cells.

**Figure 6 cancers-11-00498-f006:**
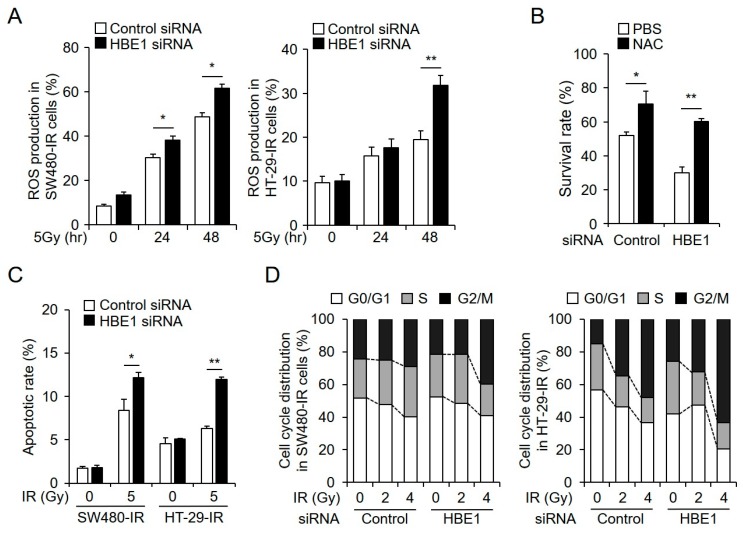
HBE1-deficiency in radiation-resistant cells increases radiation sensitivity through increased reactive oxygen species (ROS) production and cell cycle arrest. After transfecting SW480-IR and HT-29-IR cells with control or HBE1 siRNA in the presence or absence of radiation (5 Gy) exposure, ROS production and the expression of HBE1 were measured by real-time PCR. (**A**) Intracellular ROS levels were analyzed by flow cytometry using CM-H2DCFDA for the indicated times, compared to those in the corresponding controls * *p* < 0.05, ** *p* < 0.01. (**B**) Bar graph showing that HBE1 depletion via HBE1 siRNA decreases the survival rate, as determined by a clonogenic assay, after 14 days. Cells were pre-treated with the antioxidant N-acetyl cysteine (NAC). * *p* < 0.05, ** *p* < 0.01 (**C**) Ionizing radiation (IR)-resistant cells were transfected with control or HBE1 siRNA and exposed to 5 Gy irradiation. Forty-eight hours later, the apoptotic rate was determined by FACS analysis based on annexin V staining. * *p* < 0.05, ** *p* < 0.01. (**D**) After exposure to radiation (5 Gy) for 24 h, cell cycle arrest at G2/M in SW480-IR and HT-29-IR cells transfected with control or HBE1 siRNA was analyzed by propidium iodide (PI) staining. All data shown are the means ± SD of three independent experiments.

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
