# Peer review of "Epsilon-Globin HBE1 Enhances Radiotherapy Resistance by Down-Regulating BCL11A in Colorectal Cancer Cells"

_cancers, 2019, doi:10.3390/cancers11040498_

Round 1

Reviewer 1 Report

The authors describe increased HBE expressions, reduced cell viability and increased G2/M arrests, after irradiation for colorectal cancer cells and radioresistant clons. The data imply that targeting of HBE1 may increase the efficacy of potential radioresistant colorectal tumor clons. The manuscript is of interest and could be accepted after addressing the following major adjustments.

Introduction:

Nicely written. Please provide more information about the clinical implications of the subject.

Results:

- Line 87 ff. should be mentioned in the M&M section.

- How did the authors account for intermitotic cell deaths that could occur several days after irradiation.

- Figure 1a. please provide standard deviations.

-Please provide further subheadings to increase the readability.

-How and why did the authors decide to evaluate HBE1 expression?

-There are several elements in the results part that belong into the discussion section (e.g. line 148/149)

-Flow cytometry is additionally useful after 72 or 96 hours to provide more profound information concerning prolonged effects

- Why did the authors decide to use only 5Gy in the majority of measurements, since adjuvant RT or normofractionated neoadjuvant RT is performed in 1,8 – 2Gy per fraction. Furthermore, subletal damage repair is not taken into account.

Discussion

Strength and weaknesses were not discussed

M&M

Weakness: Only two replicas were performed instead of triplets.

Was the G0 cell fraction determined?

Author Response

We thank the Editor and the Reviewers for their detailed and helpful comments. We have
revised the manuscript accordingly as described in the point-by-point response below. To
this end, we have performed additional experimental work, resulting in the generation in new
or modified figure panels. We hope that that these changes, including the new experimental
work, will render the manuscript suitable for publication in Cancers.

Reviewer #1
Introduction:
Nicely written. Please provide more information about the clinical implications of the subject.

The following references have been added
[12] Khalil, A. M.; Moslemi, D.; Sadati, Z. A.; Ebrahimnezhad, G.M.; Mosapour, A.; Haghhaghighi, A.; Halalkhor, S.; Bijani, A.; Parsian, H. Pre and post radiotherapy serum oxidant/antioxidant status in breast cancer patients: Impact of age, BMI and clinical stage of the disease. Rep Pract Oncol Radiother. 2016,21,141-8. doi: 10.1016/j.rpor.2015.12.009
[13] Wozniak, A.; Masiak, R.; Szpinda, M., Mila-Kierzenkowska, C.; Woźniak, B.; Makarewicz,R.; Szpinda, A. Oxidative stress markers in prostate cancer patients after HDR brachytherapy combined with external beam radiation. Oxid Med Cell Longev. 2012, 789870. doi: 10.1155/2012/789870.

Results:
- Line 87 ff. should be mentioned in the M&M section.

Methods for establishing radiation resistant cell lines are described in method 4.2
- How did the authors account for intermitotic cell deaths that could occur several days after
irradiation.

Although the DNA damage response pathway removes much of the initial damage cause by
irradiation, they are unable to prevent some cells with DNA break or DNA rearrangements from
entering mitosis. The consequence of incomplete or improper DNA repair becomes mitotic
catastrophe that is a trigger for cell death (Nature Reviews Cancer volume 5, pages 231–237
(2005))
- Figure 1a. please provide standard deviations.
The figure 1a has been changed.
-Please provide further subheadings to increase the readability.

The following subheading have been changed
2.4 Enhancement of HBE1 levels is downregulated by the activation of JNK and ER stress signaling
-> Enrichment of HBE1 levels decreases radiation-induced ER-stress signaling
2.5 HBE1 expression is mediated by the depletion of BCL11A expression
-> HBE1 expression levels are regulated by the depletion of BCL11A
2.6 Deficiency in HBE1 expression in radiation-resistant cells increases radiation-induced cell death
-> HBE1 deficiency affects radiation sensitivity

-How and why did the authors decide to evaluate HBE1 expression?

By comparing RNA-seq data from several colorectal cancer (CRC) cell lines with differing radiation
sensitivities, we identified a candidate radiation resistance-associated gene, HBE1

-There are several elements in the results part that belong into the discussion section (e.g. line
148/149)

The following texts have been changed
Original manuscript: line 148-149, line 206-208

-Flow cytometry is additionally useful after 72 or 96 hours to provide more profound information
concerning prolonged effects

Physiological levels of reactive oxygen play critical roles in many cellular functions. Upon the
radiation damage, protective mechanisms may or may not be sufficient to cope with the stress.
Short-term biological effects occur via water radiolysis. But radiation-induced cellular oxidative
stress may be prolonged due to long-term effects on oxidative metabolism (Cancer Lett. 2012 Dec
31; 327(0): 48–60.). Therefore, we have confirmed long-term results.

- Why did the authors decide to use only 5Gy in the majority of measurements, since adjuvant RT
or normofractionated neoadjuvant RT is performed in 1,8 – 2Gy per fraction. Furthermore, subletal
damage repair is not taken into account.

The conventional radiation dose used to be approximately 40-50 Gy with 1.8~2 Gy for long-course
radiotherapy, however short-course radiotherapy was used 5Gy fraction without chemotherapy.
(Radiat Oncol J. 2017 Dec; 35(4): 295–305. doi: 10.3857/roj.2017.00395). Also, our 5Gy fraction
results are more efficient than low dose fractions
.

Discussion
Strength and weaknesses were not discussed

In response to this comment we have changed the discussion section of the manuscript.

M&M
Weakness: Only two replicas were performed instead of triplets.

The experiment was repeated several times and the same results were obtained.

Was the G0 cell fraction determined?

Propidium Iodode is possible to identify the three interphase stages of the cell-cycle by using flow
cytometry. Cells that are in the G0/G1 phase (before DNA synthesis) have a defined amount (1×)
of DNA. During S phase (DNA synthesis), cells contain between 1× and 2× DNA levels. Within the
G2 or M phases (G2/M), The cells have a 2× amount of DNA
.

Reviewer 2 Report

The authors in this in vitro study target the radioresistance of colorectal cell lines and colorectal cancer. Based on their previous work, in which and after comparing RNA-seq data from several colorectal cancer (CRC) cell lines with differing radiation sensitivities, identified a possible radiation resistance-associated gene, namely, epsilon-globin (HBE1) . The increase or overexpression of HBE1 according to the authors leads to a reduction of radiation-induced cell death, ROS and apoptosis. Although the results are consistent , the authors mechanistically speaking miss the role of DNA damage which would explain most of their findings and/or impact on DNA repair of the HBE1 over- or down-regulation. In addition, the discussion and/or introduction involving extensive bioinformatic studies on radiationresistance would only the benefit the relatively poor discussion and status of knowledge. 

Specific (not minor) comments

1. The title must read : "Expression of epsilon-globin HBE1 attenuates the transcriptional regulator BCL11A ..."?

2. Abstract must be revised and really explain their main questions first and then in order methds and main results

3.BCL11A and apoptosis (Biomed Rep. 2013 Jan-Feb; 1(1): 47–52: https://www.ncbi.nlm.nih.gov/pmc/articles/PMC3956826/) : The role of BCL11A in apoptosis has been previously described and a proper discussion must be done. 

4. The role of HBE1 as an antioxidant: This is a very risky strategy since the HBE1 can impact so many different other pathways thorught regulation of DDR damage response (see DDR correlation with other pathways of systemic responses (2015: The DNA damage response and immune signaling alliance: Is it good or bad? Nature decides when and where. Pharmacol Ther, 154, 36-56). This can explain that in tumors the HBE1 is not evident and other pathways takeover like DNA repair leading to resistance and in the organisms various systemic responses.  

Author Response

Specific (not minor) comments
1. The title must read : "Expression of epsilon-globin HBE1 attenuates the transcriptional
regulator BCL11A ..."?

We agreed to your suggestion and changed the title.

-> Epsilon-globin HBE1 enhances radiotherapy resistance by down-regulating BCL11A in
colorectal cancer cells

2. Abstract must be revised and really explain their main questions first and then in order
methods and main results

We have corrected it in the abstract section of revision.

3.BCL11A and apoptosis (Biomed Rep. 2013 Jan-Feb; 1(1): 47–52:
https://www.ncbi.nlm.nih.gov/pmc/articles/PMC3956826/) : The role of BCL11A in apoptosis
has been previously described and a proper discussion must be done.

We appreciate your insightful suggestions. We agree that it is better to provide more details
about the role of BCL11A in the Discussion section.
The following text has been added to discussion section
Furthermore, bcl11a is known as a non-autonomous T cell tumor suppressor gene. Interestingly,
another bcl11 gene family, bcl11b, acts as a tumor suppressor gene. Therefore, disruption of
bcl11b contributes to oncogenesis. Also, bcl11b-lacking cells revealed deregulation of cell
cycle checkpoint. Inappropriate cell cycle induces DNA damage accumulation and ultimately
contributes to tumor progression. On the other hand, downregulation of BCL11A is known to
be induces apoptosis in B lymphoma cell. bcl11a has also been reported to associate with a
variety of B-cell malignancies in humans. These results make it difficult to distinguish whether
BCL11A is an oncogene or a tumor suppressor gene. Thus, further investigation of bcl11a
function is required for effective radiotherapy strategy.

4. The role of HBE1 as an antioxidant: This is a very risky strategy since the HBE1 can impact
so many different other pathways thorught regulation of DDR damage response (see DDR
correlation with other pathways of systemic responses (2015: The DNA damage response and
immune signaling alliance: Is it good or bad? Nature decides when and where. Pharmacol Ther,
154, 36-56). This can explain that in tumors the HBE1 is not evident and other pathways
takeover like DNA repair leading to resistance and in the organisms various systemic responses.

Thanks for pointing us to those references, However, our results show that HBE1 reduces DNA
damage response via reducing ROS generation rather than affecting DNA repair activity. These
results appear to be resistant to radiation in HBE1 expressing CRC cells. However, we did not
discuss the relationship between antioxidants and radiation resistance.
Therefore, the following text has been added to discussion section
Exposure to ionizing radiation has been definitively linked to mitochondrial-dependent
ROS/reactive nitrogen species generation in tumor cells, and increased ROS generation in
mitochondria following low-dose ionizing radiation has been shown to contribute significantly
to radiosensitivity, and cell death. Also, a decline in tissue antioxidant is observed during the
radiotherapy for breast cancer patient. An aberration in redox status affects the host immune system. However, there are some debates about whether it is beneficial or detrimental to the
use of antioxidants. Nevertheless, the antioxidant be considered an important factor in the
response of tumors to radiation therapy. These observations are consistent with the concept that
the expression levels of hemoglobin subunits may in fact represent a more important response
to radiotherapy in CRC cells.

Reviewer 3 Report

The study by Sang Yoon Park and colleagues examined the molecular mechanisms which may provide radioresistance to colorectal cancer cells. They found that haemoglobin E1 is upregulated while its transcriptional repressor, BCL11A is downregulated in radioresistant cells. The study proposes that HBE1 provides radioresistance by inhibiting cell death signalling via reducing oxidative stress.

The experiments of the study in general are well designed and executed and the main conclusions are supported with evidence. Where the manuscript requires improvements is more careful conclusions at the level of individual experiments, inclusion of a few additional measurements to corroborate the conclusions and amendments in the discussion.

The specific comments are below:

Major comments:

1.      The study shows evidence that HBE1plays a role in reducing oxidative stress. However, the study also shows that the ROS scavenger, NAC further protects HBE1 overexpressing cells (FIG 3E), indicating that HBE1 may only protect against certain types of oxygen/free radicals, or its not able to sequester all ROS produced. To clarify this point, the authors should repeat the experiment shown in Figure 3D with including NAC treatment both in mock transfected and HBE1 overexpressing cells and determine ROS levels.

2.      The study poorly addresses the role of ER stress in irradiation-induced apoptosis and the role of HBE1 in reducing ER stress and ER stress-induced JNK activation. Firstly, characteristic markers of ER stress were not detectable, namely Grp78 induction and eIF2a phosphorylation and the detected phosphorylation of Ire1 is not supported by a total Ire1 blot. To confirm these findings, the authors would need to repeat these western blots and include detection of total Ire1 as well as a positive control sample where ER stress was induced by thapsigargin and/or tunicamycin. The levels of Ire1 and other ER stress marker activation in the positive control will show the level of ER stress induced by irradiation. Addition of spliced XBP1 mRNA expression would further support these results.

3.      Regarding the role of JNK, the authors show no connection between induction of ER stress and JNK phosphorylation. As ER stress is far from being the only trigger of JNK activation, conclusions with this regard must be toned down.

4.      The study used different doses of irradiation, which varied on an extremely broad scale. As the majority of the cells (over 90%) typically died after exposure to 5Gy of irrad. Why was it decided to expose the cells to 120 Gy? These experiments would need to be replaced with studies using a dose range between 1-5 Gy.

Minor comments:

1.      The viability graphs in Figure 1 would need to have the Y axis changed from logarithmic to linear to better show the differences between the resistant vs sensitive cell lines.

2.      The discussion requires a substantial modification. The section on hypoxia and anemia is very unclear and mostly irrelevant to the study. The authors did not study hypoxia, and made no indications that hypoxia would play a role in radioresistance.

3.      With regards to anemia, the current literature refers to systemic anemia, i.e. reduced blood cell production, as opposed to low or high haemoglobin production in/by tumours. The relevance of this section in discussing the results is unclear.

4.      Figure 5d. The legend I assume should state that cells were transfected with BCL11A, not HBE1.

5.      In row 215 the wording of the sentence stating that ER stress-induced apoptosis triggers JNK activation is incorrect. I assume it meant to state that prolonged ER stress leads to JNK activation triggering cell death.

6.      The wording of the legend for Figure 2b has a grammatical error (row 157-158).

7.      Please add to all experiments the time of the treatment (how many hours/days of irradiation, etc).

Author Response

Major comments:
1. The study shows evidence that HBE1 plays a role in reducing oxidative stress.
However, the study also shows that the ROS scavenger, NAC further protects HBE1
overexpressing cells (FIG 3E), indicating that HBE1 may only protect against certain types of
oxygen/free radicals, or its not able to sequester all ROS produced. To clarify this point, the
authors should repeat the experiment shown in Figure 3D with including NAC treatment both
in mock transfected and HBE1 overexpressing cells and determine ROS levels.

Cells were treated with 10mM NAC for 1 hr. The intracellular ROS level is measured by flow
cytometry at 48 hr after irradiation (5Gy) exposure. The results were attached to supplementary
data. (figure S1). There was no significant difference in ROS levels between Mock and HBE1
expressing cells. However, the difference is expected after enough time has passed. Because
the generation of ROS by irradiation has been increasing over time. Also, the clonogenic assays
are cultured for 14 days after exposure to radiation. However, NAC is not sufficient because it
has a half-life of about 5.6 hours. (Getting a Knack for NAC: N-Acetyl-Cysteine. Innov Clin
Neurosci. 2011, 8, 10–14). The differences in Figure 3.E are also due to these differences.

2. The study poorly addresses the role of ER stress in irradiation-induced apoptosis and
the role of HBE1 in reducing ER stress and ER stress-induced JNK activation. Firstly,
characteristic markers of ER stress were not detectable, namely Grp78 induction and eIF2a
phosphorylation and the detected phosphorylation of Ire1 is not supported by a total Ire1 blot.
To confirm these findings, the authors would need to repeat these western blots and include
detection of total Ire1 as well as a positive control sample where ER stress was induced by
thapsigargin and/or tunicamycin. The levels of Ire1 and other ER stress marker activation in
the positive control will show the level of ER stress induced by irradiation. Addition of spliced
XBP1 mRNA expression would further support these results.

Thank you for pointing out the missing parts of our experimental results. We accepted your
advice and added total ire1a blot to support phosphorylation of Ire1. Also, after treatment with
Thapsigargin as an ER stress positive control, the results were attached to supplementary data.
(figure S2). Additionally, RT-PCR was used to confirm the increase in spliced XBP-1 mRNA,
in order to demonstrate a decrease in radiation-induced ER-stress in HBE1 expressing cells.

3. Regarding the role of JNK, the authors show no connection between induction of ER
stress and JNK phosphorylation. As ER stress is far from being the only trigger of JNK
activation, conclusions with this regard must be toned down.

Our results are based on the fact that the IRE1 pathway induces apoptosis through the JNKmediated signaling pathway. But as you pointed out, this argument has not been proven
sufficiently, so we have modified some of the sentences.

4. The study used different doses of irradiation, which varied on an extremely broad
scale. As the majority of the cells (over 90%) typically died after exposure to 5Gy of irrad.
Why was it decided to expose the cells to 120 Gy? These experiments would need to be
replaced with studies using a dose range between 1-5 Gy.

There was no difference in the expression of HBE1 in parental cells when exposed to low grade
radiation. However, the increased expression of HBE1 appeared when exposed to the same
total dose (120Gy) which used to generate radiation resistant cell lines.

Minor comments:
1. The viability graphs in Figure 1 would need to have the Y axis changed from
logarithmic to linear to better show the differences between the resistant vs sensitive cell lines.

As requested, the figure 1a has been changed.

2. The discussion requires a substantial modification. The section on hypoxia and
anemia is very unclear and mostly irrelevant to the study. The authors did not study hypoxia,
and made no indications that hypoxia would play a role in radioresistance.

We removed the sentence from the manuscript.

3. With regards to anemia, the current literature refers to systemic anemia, i.e. reduced
blood cell production, as opposed to low or high haemoglobin production in/by tumours. The
relevance of this section in discussing the results is unclear.

We sincerely appreciate your insightful comments as you suggested, we have corrected

4. Figure 5d. The legend I assume should state that cells were transfected with BCL11A,
not HBE1.

We are very sorry for this mistake, we have corrected

5. In row 215 the wording of the sentence stating that ER stress-induced apoptosis
triggers JNK activation is incorrect. I assume it meant to state that prolonged ER stress leads
to JNK activation triggering cell death.

Thank you for noting this. We have corrected it

6. The wording of the legend for Figure 2b has a grammatical error (row 157-158).

This sentence has been revised.

7. Please add to all experiments the time of the treatment (how many hours/days of
irradiation, etc).

We agree that it is better to provide more details about the experimental conditions. we have
corrected it in the revision.

Round 2

Reviewer 1 Report

Accept

Reviewer 3 Report

No further comments